# *Toxoplasma gondii* Seropositivity and Co-Infection with TORCH Complex Pathogens in Pregnant Women from Araçatuba, Brazil

**DOI:** 10.3390/microorganisms12091844

**Published:** 2024-09-06

**Authors:** Sabrina Santos Firmino, Thaís Rabelo Santos-Doni, Vitória Maria Farias Silva, Aressa Cassemiro Micheleto, Ma Scalise de Souza, Bruna Lima Hortêncio, Aline do Nascimento Benitez, Yasmin Melim Bento, Gabriele Zaine Teixeira Debortoli, Jancarlo Ferreira Gomes, Italmar Teodorico Navarro, Katia Denise Saraiva Bresciani

**Affiliations:** 1Faculdade de Medicina Veterinária, Universidade Estadual Paulista (UNESP), Araçatuba 16050-680, São Paulo, Brazil; ss.firmino@unesp.br (S.S.F.); vmf.silva@unesp.br (V.M.F.S.); aressa.micheleto@unesp.br (A.C.M.); scalise.souza@unesp.br (M.S.d.S.); brunalima9898@gmail.com (B.L.H.); 31benitez.aline@gmail.com (A.d.N.B.); yasminmb.vet@gmail.com (Y.M.B.); 2Instituto de Ciências Agrárias (ICA), Universidade Federal dos Vales do Jequitinhonha e Mucuri (UFVJM), Avenida Universitária, Unaí 38610-000, Minas Gerais, Brazil; gabriele.zaine@ufvjm.edu.br; 3Faculdade de Ciências Médicas e Instituto de Computação, Universidade Estadual de Campinas, Campinas 13083-887, São Paulo, Brazil; jgomes@ic.unicamp.br; 4Departamento de Medicina Veterinária Preventiva, Universidade Estadual de Londrina (UEL), Londrina 86057-970, Paraná, Brazil; italmar@uel.br

**Keywords:** toxoplasmosis, prevalence, congenital syndromes, prenatal screening, epidemiology, congenital infections, public health

## Abstract

This study examined the seropositivity of *T. gondii* and coinfections with other TORCH pathogens among pregnant women attending 17 Basic Health Units (UBS) in Araçatuba, SP, Brazil. Of the 711 pregnant women seen at these UBS, only 297 were tested for *T. gondii*. Of the women tested for *T. gondii* (*n* = 297), 26.9% had IgG antibodies, 6.7% had IgM, and 32.0% tested positive for either or both. Only 1.4% showed both IgG and IgM antibodies, while 67.7% were non-reactive. The seropositivity was 17.1% for syphilis, 63.2% for rubella, 0.9% for hepatitis C, 0.9% for dengue, 17.9% for COVID-19, and 0.9% for herpes simplex (types 1/2). Coinfections with syphilis, rubella, and herpes simplex were also noted. Higher education levels appeared to protect against *T. gondii* seropositivity. The findings highlight a significant prevalence of *T. gondii* among pregnant women, with variation across UBSs, pointing to socioeconomic, behavioral, and environmental factors as influential. We also observed co-occurrence with other infections, such as syphilis, rubella, and herpes simplex. The study underscores the need for targeted public health interventions to reduce the risks of congenital infections.

## 1. Introduction

Toxoplasmosis is caused by the protozoan *Toxoplasma gondii* and is part of the TORCH complex, a group of infections that can lead to serious congenital problems when acquired during pregnancy [1,2]. TORCH complex infections include toxoplasmosis (TOX), others (syphilis, hepatitis, varicella-zoster, HIV), rubella (RV), cytomegalovirus (CMV), and herpes simplex (HSV-1/HSV-2) [3].

Infection by *T. gondii* generally occurs through the ingestion of sporulated oocysts, eliminated by cats, which are its definitive hosts, contaminating the environment, food, or water. Consumption of raw or undercooked meat containing parasite cysts is another common route of infection. Typically, the infection is asymptomatic and self-limiting [4]. This protozoan can infect virtually all warm-blooded animals, including humans [5].

During pregnancy, the infection is particularly serious, as it can be transmitted vertically to the fetus, resulting in congenital toxoplasmosis [6]. The initial infection in humans is often subclinical, but the parasite can persist in the body in a latent form, reactivating in situations of immunosuppression [7].

Congenital toxoplasmosis occurs when the infection is transmitted from the mother to the fetus through the placenta. The consequences of this infection can vary depending on the timing of transmission during pregnancy [8,9]. Infections early in pregnancy tend to result in more severe consequences, such as spontaneous abortion, fetal death, or serious neurological damage to the fetus, including hydrocephalus, intracranial calcifications, and chorioretinitis, which can lead to blindness. Infections acquired later in pregnancy may result in milder or even asymptomatic conditions at birth, with clinical manifestations appearing months or years later, such as hearing problems and delays in neuropsychomotor development [10,11,12,13,14,15]. 

In developed countries like the U.S.A., about 10% of pregnant women who are infected exhibit symptoms, which are often nonspecific [7]. In contrast, the prevalence of toxoplasmosis among pregnant women in Brazil ranges from 50% to 80% [16]. A meta-analysis estimated a pooled seroprevalence of *T. gondii* infection among pregnant women at 40.0% (95% CI, 37.0–44.0%), highlighting a significant level of exposure in this demographic group [17]. According to the meta-analysis by Bigna et al. [18], the global IgM seroprevalence of *T. gondii* infection was 1.9% (95% CI: 1.7–2.3), and the global IgG seroprevalence was 32.9% (95% CI: 29.4–36.4). Among the WHO regions, the Americas had the highest IgG seroprevalence at 45.2% (95% CI: 33.4–53.4), indicating a significant exposure level in this region.

Coinfection with other TORCH complex pathogens, such as syphilis, hepatitis C, dengue virus, COVID-19, rubella, CMV, and herpes simplex, poses additional risks for pregnant women and their fetuses [19,20]. These infections can lead to a range of congenital anomalies and complications, making it imperative to understand their combined impact. Comprehensive epidemiological analyses are necessary to inform public health interventions and develop specific strategies to reduce the incidence and impact of these congenital infections [2].

This study aimed to investigate the seropositivity of *T. gondii* and the rates of coinfection with other TORCH complex pathogens among pregnant women in Araçatuba, SP, Brazil. By analyzing serological data and examining associated risk factors, it is expected to provide relevant information about these infections and contribute to the development of effective public health strategies to reduce the morbidity and mortality associated with congenital infections.

## 2. Materials and Methods

### 2.1. Design and Study Area

This cross-sectional, descriptive, and quantitative study was conducted after approval by the Human Research Ethics Committee of the São Paulo State University (UNESP), under Opinion N° 2.625.160. The research was carried out in the municipality of Araçatuba, located in the northwest of the State of São Paulo (21°12′32′′ S, 50°25′58′′ W), at an altitude of 380 m above sea level, supplied by the Tietê and Ribeirão Baguaçu rivers, and characterized by a semi-humid tropical climate. In 2022, the municipality’s population was estimated at 200,124 inhabitants, with an infant mortality rate of 13.91 deaths per thousand live births.

### 2.2. Context of Primary Health Care

The municipality has a basic health care network consisting of 20 service units, with 17 located in urban areas and 3 in rural areas. Three of these units do not provide prenatal care and instead refer pregnant women to the nearest facilities. For this study, 17 units were selected, including 16 in urban areas and 1 in a rural area.

### 2.3. Data Collection

The analysis was carried out in partnership with the Municipal Health Department (SMS) to obtain data on pregnant women treated at Basic Health Units (BHU). In total, 711 pregnant women were treated at the 17 selected BHUs from February 2022 to March 2023. The distribution of pregnant women per unit was as follows: BHU 0 (43), BHU 1 (35), BHU 2 (21), BHU 3 (76), BHU 4 (35), BHU 5 (35), BHU 6 (27), BHU 7 (26), BHU 8 (70), BHU 9 (30), BHU 10 (20), BHU 11 (51), BHU 12 (122), BHU 13 (34), BHU 14 (40), BHU 15 (51), BHU 16 (26), and BHU 17 (4). 

Weekly visits were made to the BHU, determined by neighborhoods, for in-person data collection. In this way, the collection was comprehensive and thorough, which was essential for detailed analysis and for preparing the study.

### 2.4. Data Collection Instrument

Epidemiological data were collected through a transcription of medical records used by health units, using all the information necessary for analysis. The information collected included race, education, basic sanitation conditions, marital status, pregnancy planning and acceptance, high risk and reason for high risk, and IgG and IgM results for *T. gondii*, syphilis, rubella, COVID-19, and any other TORCH complex disease. To safeguard participant identities, we implemented stringent anonymization procedures. Each participant was assigned a unique code, and all personally identifiable information was either removed or encrypted in the datasets. Additionally, access to the data was strictly limited to authorized personnel, ensuring confidentiality and adherence to ethical standards.

### 2.5. Laboratory Analyses

The laboratory tests were carried out at the Mahatma Gandhi Clinical Analysis Laboratory, located in Araçatuba, which maintains a partnership with the Municipal Government of Araçatuba. All examinations followed the protocols established by the Ministry of Health [21].

The Microparticle Chemiluminescent Immunoassay (CMIA) system was used to detect antibodies of *T. gondii* (IgM/IgG) and syphilis (Abbott Ireland, Diagnostics Division, Sligo, Ireland), following the manufacturer’s instructions. Confirmation of syphilis in non-treponemal reactive results was confirmed with a treponemal test, the fluorescent treponemal antibody absorption assay (FTA-ABS), which detects serum antibodies specific for *Treponema pallidum* (WAMA Diagnostica, São Paulo, Brazil). 

Electrochemiluminescence (ECLIA) was used for the diagnosis of rubella (IgM/IgG) and hepatitis C (IgG/IgM) (Roche Diagnostics, Switzerland), while chemiluminescence (CLIA) was used for the detection of herpes simplex type 1 and 2 (IgM/IgG) using kits from Euroimmun (Lübeck, Germany). The diagnostic methods used for dengue and COVID-19 were, respectively, NS1 antigen test and rapid antigen test (RT-Ag) (WAMA Diagnostica, São Paulo, Brazil). All analyses were performed strictly in accordance with the manufacturers’ instructions.

### 2.6. Data Analysis

Statistical tests were performed using STATA/SE software (version 16.1; Stata Corp LLC, College Station, TX, USA). We used descriptive statistics to characterize the demographic variables that were described in terms of percentages, standard errors, and 95% confidence intervals. Data collected from 711 pregnant women in 17 BHUs were analyzed to determine seropositivity of *T. gondii* and rates of co-infection with TORCH pathogens in pregnant women.

For inferential statistics, the dependent variable was seropositivity for IgG and/or IgM against *T. gondii*, while the independent variables included education, race, basic sanitation, pregnancy planning, pregnancy acceptance, and high-risk pregnancy [22]. To explore possible interactions between independent variables and assess whether the effect of one variable on *T. gondii* seropositivity is modified by another variable, we conducted binary logistic regression analyses with interaction terms. Due to the overrepresentation of urban health units in our study, with only one unit located in a rural area, a meaningful comparison of *T. gondii* seropositivity between urban and rural areas was not feasible. A *p*-value less than 0.05 was considered statistically significant.

The georeferencing (Figure 1) of the visited BHUs was performed using a GPS device (GPS Garmin eTrex 30). The data were then transported to the QGIS desktop program (3.26.3) [23], where they were placed on the map of Araçatuba/SP. The database modeling and graph plotting steps were performed at the Animal Parasitology and Zoonoses Laboratory of the Institute of Agricultural Sciences, Federal University of the Jequitinhonha and Mucuri Valleys.

## 3. Results

Table 1 shows the distribution of pregnant women attended prenatally and the seropositivity of *T. gondii* in 17 Basic Health Units (BHU) in Araçatuba, SP, Brazil.

The data presented in this table indicate that, among the 711 pregnant women attended to in 17 BHUs in Araçatuba/SP, only 297 showed results for *T. gondii*.

Among the 297 pregnant women evaluated, 80 (26.9%) had IgG antibodies to *T. gondii*, indicating previous exposure to the parasite. Additionally, 20 pregnant women (6.7%) were identified with IgM antibodies to *T. gondii*. In the combined analysis of IgG or IgM, 95 of the pregnant women (32.0%) were positive for one or both types of antibodies. Simultaneous detection of IgG and IgM occurred in only 4 (1.4%) pregnant women, and 201 (67.7%) pregnant women presented non-reactive IgG and IgM serology, revealing themselves to be susceptible to acquiring the infection.

Data on *T. gondii* seropositivity among pregnant women in the BHUs of Araçatuba, SP, show variability, which can be attributed to socioeconomic, behavioral and environmental factors. While BUH 13, together with BHU 16 and BHU 17, did not register positive cases among the pregnant women treated, other units presented high infection rates. BHUs 14 and 7 stood out with the highest rates, reaching 50% and 46.7%, respectively, followed by BHU 11 with 42.9% and BHU 12 with 39.4%.

This data can be viewed in detail in Figure 1. The map geographically represents the seropositivity of toxoplasmosis among pregnant women treated at 17 Basic Health Units (BHU) in Araçatuba, São Paulo.

Table 2 shows the demographic characteristics of pregnant women seropositive for *T. gondii* who attended prenatal care in 17 BHUs in Araçatuba, SP, Brazil.

The average age of the participants was 26.2 years. Regarding the level of education, the largest percentage of women (39.4%) completed high school, while 27.6% of women did not respond about their education. Regarding ethnicity, most women did not respond (58.3%), but among those who responded, 23.6% were white and 14.5% were brown.

In terms of basic sanitation, 38.4% considered themselves to have access to adequate sanitation, but 61.3% did not answer this question. Regarding pregnancy planning, 40.7% of women said they did not plan the pregnancy, 26.3% planned it, and 33.0% did not respond. Acceptance of pregnancy shows that 61.6% of women accepted the pregnancy, while 3.0% did not, and 35.4% did not respond. Finally, 25.9% of women were in a high-risk pregnancy, while the majority (74.0%) were not high-risk.

Table 3 shows the univariate logistic regression of *T. gondii* seropositivity according to the predictors evaluated in pregnant women in Araçatuba/SP.

The education variable has an odds ratio of 0.78, which suggests that increases in the level of education are associated with a decrease in the probability of being seropositive for *T. gondii*. The 95% confidence interval for this odds ratio ranges from 0.62 to 0.98, and the *p*-value of 0.033 indicates that this effect is statistically significant at the 5% level.

This model suggests that there is no significant difference in seropositivity for *T. gondii* (IgG/IgM) among the other variables analyzed, such as race, basic sanitation, women with planned pregnancies, acceptance of pregnancy, and presence of high risk.

Table 4 presents the seropositivity of the diseases analyzed and correlations between the presence of antibodies for toxoplasmosis (IgG/IgM) and other infectious diseases in pregnant women, using Spearman’s correlation coefficient (rho) and the respective *p*-values to assess the statistical significance of these correlations among pregnant women who received prenatal care in 17 Basic Health Units (BHU) in Araçatuba/SP, Brazil. 

The data reveal that, of the 711 women monitored, 297 underwent tests to detect *T. gondii* antibodies (IgG/IgM), while only 117 were evaluated for other diseases such as syphilis, rubella, hepatitis C, dengue virus, COVID-19, and herpes simplex.

Of these, rubella showed the highest seroprevalence, with 63.2% (74/117), followed by COVID-19 with 17.9% (21 of 117). Seropositivity for *T. gondii* shows that 32.0% of the pregnant women evaluated were seropositive for IgG/IgM. Notably, some diseases show co-occurrence with toxoplasmosis. The non-significant correlations with toxoplasmosis indicate that, despite some weak associations, there is no strong evidence that the presence of antibodies to toxoplasmosis is directly related to other infectious diseases in the sample studied.

Data analysis revealed a significant negative correlation between rubella and syphilis seropositivity in pregnant women, with a Spearman coefficient (rho) of −0.5387 and a significantly low *p* value (*p* < 0.001). This result suggests that pregnant women who are seropositive for rubella tend to be less likely to be seropositive for syphilis.

## 4. Discussion

The prevalence of toxoplasmosis in pregnant women varies considerably between different regions of Brazil, as indicated by several studies. This study identified a prevalence of anti-*T. gondii* antibodies (IgG or IgM) of around 32.0%, which is lower than the seropositivity reported in the North region, city of Gurupi, TO, with a high prevalence (IgG or IgM) of 68.4% (333/487) among pregnant women [16].

The prevalence of maternal chronic infection (IgG) was 26.9% (80/297); similar results were described by Pereira et al. [24] in Presidente Prudente/SP, with 24.6% (69/280), and by Ferreira et al. [25] with 20.9% (29/139), in Campina Grande/PB. Higher prevalence rates were reported by Figueiró–Filho et al. [26], with 91.6% (29,781/32,512) in the state of Mato Grosso do Sul; by Areal and Miranda [27], with 73.5% (847/1,153) in Vitória/ES; with Gontijo da Silva, Clare Vinaud, and de Castro [16], who reported a prevalence of 63.0% (307/487) in Gurupi/TO; by Castilho–Pelloso et al. [28], with 65.2% (10,882/16,686) in the northwest of the state of Paraná; by Santos et al. [29], with 62.5% (125/280) in Rio Grande/RS; and by Freitas et al. [30], with 59.0% (210/356) in Jaçanã/RN.

Although such variation may be related to the technique used for IgG detection, dietary habits and water sources may play a role in transmission, as previously reported [31].

Infection of pregnant women was diagnosed in 20 (6.7%) 265 cases during the first trimester of gestational age by detecting IgM antibodies, only. Indeed we did not determine IgA levels against T. gondii or assess IgG avidity in IgM-positive females. Similar results were presented by Gontijo da Silva, Clare Vinaud, and de Castro [16], who reported a prevalence of 4.9 (24/487) in Gurupi/TO. Lower results were reported by Castilho–Pelloso, Falavigna, and Falavigna–Guilherme [28], with 0.2% (26/16.686) in the northwest of the state of Paraná; by Figueiró-Filho, Lopes, Senefonte, Souza Júnior, Botelho, Figueiredo, and Duarte [26], with 0.4% (137/32,512) in the state of Mato Grosso do Sul; and by Areal and Miranda [27], with 1.3% (15/1153) in Vitória/ES. In the context of study limitations, it is important to note that we did not determine IgA levels against *T. gondii* or assess IgG avidity in IgM-positive females.

Recent worldwide systematic review studies have reported global seropositivity of anti-*T. gondii* antibodies in pregnant women; with an IgG of 32.9% and an IgM of 1.9% [32], Rostami, et al. [32] found a 33.8% global prevalence of latent toxoplasmosis in pregnant women.

Ordinal logistic regression revealed that having a higher educational level was a protective factor against seropositivity for toxoplasmosis, corroborating other studies on *T. gondii* in pregnant women in Brazil [16,33,34] and other countries [35,36,37]. Pregnant women with a higher level of education were associated with having knowledge related to toxoplasmosis, which may increase their awareness and understanding of the importance of hygiene habits to prevent diseases.

The analysis of the distribution of pregnant women attended prenatally and the seropositivity of *T. gondii* in 17 Basic Health Units (BHU) in Araçatuba, SP, Brazil, reveals fundamental data for understanding the prevalence of toxoplasmosis and identifying areas that require specific interventions. The variability in the distribution of pregnant women between the BHUs, ranging from 2.8% (BHU 14 and BHU 3) to 17.2% (BHU 12), reflects the disparity in demand and service capacity of the health units. This discrepancy may indicate an unequal distribution of resources and accessibility to prenatal health services, suggesting the need for more equitable allocation and improvement in health infrastructure [38]. Identifying these critical areas allows for more effective allocation of health resources.

TORCH infections are prevalent globally and are a major cause of increased prenatal, perinatal, and postnatal infant morbidity and mortality [39,40,41].

The syphilis positivity rate in this study was 17.1%, within the values found worldwide by Newman et al. [42], in which the estimated number of infected pregnant women by region was 535,203 in Africa (39.3%), 106,500 in the Americas (7.8%), 603,293 in Asia (44.3%), 21,602 in Europe (1.6%), 40,062 in the Mediterranean (3.0%), and 53,825 (4.0%) in the Pacific. There was also no statistically significant association observed between the presence of syphilis and *T. gondii* in the pregnant women evaluated.

The incidence of congenital rubella syndrome has been decreasing worldwide due to increased rubella vaccination coverage, but it remains a threatening and costly disease in regions where pregnant women are not immunized and do not have protective levels of IgG against rubella virus. In this survey, the overall rate of immunity against rubella among pregnant women was 63.2% (74/117), and similar results were reported by de Melo Inagaki et al. [43], who found 64.3% in Sergipe. Higher seropositivity among women was described at 98.0% in Iowa (USA) [44], 93.0% in Cartagena (Colombia) [45], 97.0% in Brazil [46], 99.5% in Türkiye [47], and 85.2% in India [48]. 

No pregnant women were detected with acute rubella infection, but 36.8% of women were susceptible to the virus, which reinforces the need for rubella vaccination campaigns to achieve greater coverage and prevent the emergence of congenital rubella syndrome. 

The absence of universal prenatal screening for the diagnosis of hepatitis C, with recommendations restricted only to women with risk factors, makes it difficult to accurately estimate its prevalence in the global population of pregnant women. However, in this survey, the overall prevalence of hepatitis C was 0.9%, in line with the results of other studies conducted in Serbia (0.9%) [49], Poland (0.9%) [50], and Canada (0.8%) [51]. In countries like Türkiye (0.07%) [52], Slovenia (0.09%) [53], and Saudi Arabia (0.05%) [54], positivity rates were lower. In contrast, significantly higher prevalences were reported in Yemen (6.0%) [55], India (2.3%) [56], and Ghana (3.4%) [57]. No co-infection of hepatitis C and *T. gondii* was observed in the pregnant women evaluated.

Infection with herpes simplex types 1 and 2 during pregnancy can result in transmission of the virus to the newborn during delivery. This study revealed a positivity rate of 0.9% (1/117), with a case of coinfection by *T. gondii*. Rates of herpes in pregnant women vary around the world. In Belgium, an incidence of 0.1% of cases was found [58]. In Norway, Eskild et al. [59] identified a frequency of 4% for acute cases of herpes infection.

The co-occurrence of infections with *T. gondii* in cases of rubella (20 cases), syphilis (6 cases), and COVID-19 (4 cases) highlights the complexity of the infectious landscape in pregnant women and the need for integrated strategies to monitor and treat multiple infections simultaneously. The concomitant presence of *T. gondii* with other TORCH complex infections may increase the risks to maternal and fetal health, requiring strict monitoring and immediate therapeutic interventions. While the absence of co-occurrence with hepatitis C, dengue, and herpes simplex is positive, surveillance must continue to ensure that these infections remain under control [3,20].

Complementing seropositivity data with stratified estimates across relevant demographic groups, such as age cohorts and geographic areas, is essential for targeting interventions and assessing program impacts. The limitations of these estimates highlight the urgent need for improved data through stronger national surveillance and monitoring systems. We also understand that resources used in testing for rubella should be allocated to vaccinating susceptible women.

Additional studies on the incidence and prevalence of vertically transmitted infectious diseases, as well as cost–benefit analyses of laboratory tests, are needed to develop effective protocols in Brazil. The lack of standardization in filling out medical records compromises the quality of the data collected, highlighting the need for constant training for health professionals. Improving data collection and record keeping by implementing standardized and consistent systems is crucial for the accurate monitoring of maternal and child health, contributing to reducing morbidity and mortality associated with congenital infections.

## 5. Conclusions

The results of this study demonstrate significant seropositivity of *T. gondii* among pregnant women treated at Basic Health Units (BHU) in Araçatuba, SP. This prevalence varies considerably between different BHUs, indicating that socioeconomic, behavioral, and environmental factors may influence exposure to the parasite. Co-occurrence of other infectious diseases, such as syphilis, rubella, and herpes simplex (1 and 2), was observed. These results are essential to guide local public health policies, focusing on educational and preventive strategies that consider the particularities of each BHU. The high prevalence of diseases belonging to the TORCH complex, especially congenital toxoplasmosis, points to the need for more targeted actions, such as awareness campaigns on infection prevention and continuous monitoring of pregnant women.

## Figures and Tables

**Figure 1 microorganisms-12-01844-f001:**
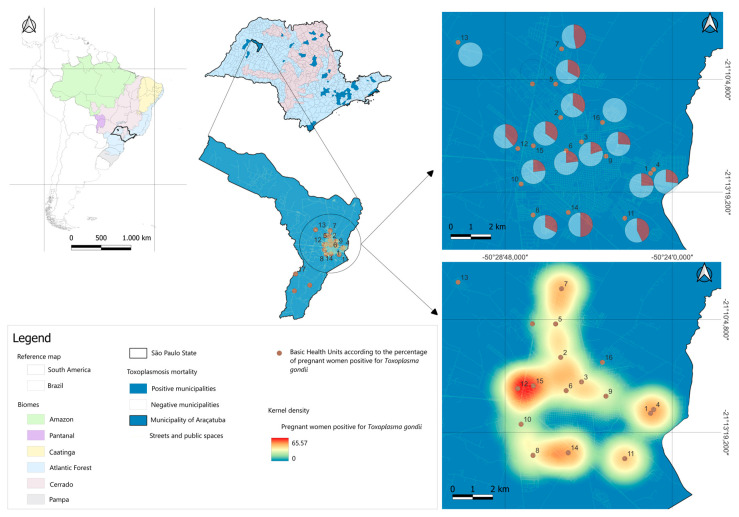
Distribution of toxoplasmosis seropositivity in pregnant women in Basic Health Units in Araçatuba, SP.

**Table 1 microorganisms-12-01844-t001:** Number of pregnant women attended prenatally and tested positive for *T. gondii* in 17 Basic Health Units (BHU) monitored in Araçatuba/SP, Brazil.

BHU	Pregnant Women			Seropositivity—*T. gondii*		Total
	%	n		%	n	
1	6.1	43		25.0	6	24
2	4.9	35		34.8	8	23
3	3.0	21		20.0	1	5
4	10.7	76		27.1	13	48
5	4.9	35		33.3	6	18
6	3.8	27		23.1	3	13
7	3.7	26		46.7	7	15
8	9.9	70		31.6	12	38
9	4.2	30		26.1	6	23
10	2.8	20		23.1	3	13
11	7.2	51		42.9	9	21
12	17.2	122		39.4	13	33
13	4.8	34		0	0	1
14	5.6	40		50.0	1	2
15	7.2	51		35.0	7	20
16	3.7	26		0	0	0
17	0.6	4		0	0	0
Total	100.0	711		32.0	95	297

**Table 2 microorganisms-12-01844-t002:** Demographic characteristics of pregnant women seropositive for *T. gondii* receiving prenatal care at 17 Basic Health Units (BHU) in Araçatuba/SP, Brazil.

Features		n (%) or Average		SE ^1^	IC95% ^2^		
Ages (average)		26.2		0.4	25.5	-	26.9
Education							
	Illiterate	0	0.0	0.0	0.0	-	0.0
	Incomplete elementary education	8	2.7	0.9	1.2	-	5.2
	Completed elementary education	50	16.8	2.2	12.8	-	21.6
	Incomplete high school education	9	3.0	1.0	1.4	-	5.7
	High school completed	117	39.4	2.8	33.8	-	45.2
	Incomplete graduation	8	2.7	0.9	1.2	-	5.2
	Completed degree	22	7.4	1.5	4.7	-	11.0
	Postgraduate studies	1	0.3	0.3	0.0	-	1.9
	Not respond	82	27.6	2.6	22.6	-	33.1
Race							
	Brown	43	14.5	2.0	10.7	-	19.0
	White	70	23.6	2.5	18.9	-	28.8
	Black	8	2.7	0.9	1.2	-	5.2
	Asian	3	1.0	0.6	0.2	-	2.9
	Not respond	173	58.3	2.9	52.4	-	63.9
Basic sanitation							
	Adequate	114	38.4	2.8	32.8	-	44.2
	Settlements	1	0.3	0.3	0.0	-	1.9
	Not respond	182	61.3	2.8	55.5	-	66.9
Planned pregnancy							
	No	121	40.7	2.8	32.8	-	44.2
	Yes	78	26.3	0.3	0.0	-	1.9
	Not respond	98	33.0	2.8	55.5	-	66.9
Acceptance of pregnancy							
	No	9	3.0	1.0	1.4	-	5.7
	Yes	183	61.6	2.8	55.8	-	67.2
	Not respond	105	35.4	2.8	29.9	-	41.1
High Risk Pregnancy							
	No	220	74.04	2.5	68.7	-	79.0
	Yes	77	25.93	2.5	21.0	-	31.3

SE ^1^: standard error; IC95% ^2^: 95% confidence interval.

**Table 3 microorganisms-12-01844-t003:** Univariate logistic regression of *T. gondii* seropositivity according to risk factors assessed in pregnant women in Araçatuba/SP.

Predictors		Seropositivity		OR ^1^	CI 95% ^2^			*p*
		n	%					
Education								
	Illiterate	0	0.0	0.78	0.62	-	0.98	0.033
	Incomplete elementary education	8	2.7					
	Completed elementary education	50	16.8					
	Incomplete high school education	9	3.0					
	High school completed	117	39.4					
	Incomplete graduation	8	2.7					
	Completed degree	22	7.4					
	Postgraduate studies	1	0.3					
	Not respond	82	27.6					
Race								
	Brown	43	14.5	1.25	0.69	-	2.28	0.45
	White	70	23.6					
	Black	8	2.7					
	Asian	3	1.0					
	Not respond	173	58.3					
Basic sanitation								
	Adequate	114	38.4	2.03 × 10^−6^	0	-	-	0.990
	Settlements	1	0.3					
	Not respond	182	61.3					
Planned pregnancy								
	No	121	40.7	1.35	0.73	-	2.50	0.331
	Yes	78	26.3					
	Not respond	98	33.0					
Acceptance of pregnancy								
	No	9	3.0	1.58	0.32	-	7.86	0.574
	Yes	183	61.6					
	Not respond	105	35.4					
High Risk Pregnancy								
	No	220	74.04	1.31	0.76	-	2.25	0.339
	Yes	77	25.93					

OR ^1^: Odds ratio; IC95% ^2^: 95% confidence interval.

**Table 4 microorganisms-12-01844-t004:** Seropositivity of the diseases analyzed and Spearman’s correlation coefficient (rho) among pregnant women from 17 Basic Health Units (BHU) in Araçatuba/SP, Brazil.

Illness	Seropositivity		Total	CI 95% ^1^			Co-Occurrence with Toxoplasma	Rho/*p*
	%	n						
*T. gondii-*IgG/IgM	32.0	95	297	26.7	-	37.3	-	
*T. gondii-*IgG	26.9	80	297	21.9	-	32.0	-	
*T. gondii-*IgM	6.7	20	297	3.9	-	9.6	-	
Syphilis	17.1	20	117	10.3	-	23.9	6	0.11/0.28
Rubella	63.2	74	117	54.5	-	72.0	20	−0.08/0.43
Hepatitis C	0.9	1	117	−0.8	-	2.5	0	-
Dengue virus	0.9	1	117	−0.8	-	2.5	0	−0.08/0.46
COVID-19	17.9	21	117	11.0	-	24.9	4	−0.04/0.67
Herpes simplex (1 and 2)	0.9	1	117	−0.8	-	2.5	1	0.14/0.19

**^1^** CI: confidence interval.

## Data Availability

The raw data supporting the conclusions of this article will be made available by the authors on request.

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
