# Peer review of "Toxoplasma gondii Seropositivity and Co-Infection with TORCH Complex Pathogens in Pregnant Women from Araçatuba, Brazil"

_microorganisms, 2024, doi:10.3390/microorganisms12091844_

Round 1
Reviewer 1 Report
Comments and Suggestions for Authors
Thank you for the opportunity to review this manuscript.
Abstract: please write “T. gondii” in Italic throughout the manuscript.
Introduction: I would personally add a paragraph on serology for the diagnosis of acute/chronic toxoplasmosis.
Materials and methods: “2.5 Análise laboratorial” does not sound English to me ☹
Please add the assay and the reagent for all the laboratory tests included in the study.
Results:
Line 154: “suggesting recent infection” I would delete this statement without an additional IgA antibody against T. gondii test and an IgG avidity test.
Please see: https://www.ncbi.nlm.nih.gov/pmc/articles/PMC4609698/
I don’t see anywhere in the manuscript a comparison between urban and rural areas, since T. gondii seropositivity is linked to rural areas, and you know the location of each health unit, please include this comparison as well. Or, if for some reason you don’t want to do it, maybe due to the overrepresentation of the urban regions, mention it in the newly limitation section 😊.
Discussion:
Please add the study limitations; maybe not determining IgA against T. gondii and IgG avidity in IgM positive females could be considered one.
Line 245: “acute infection (IgM) in pregnant women was diagnosed in 20 (6.7%) cases during 245 the first trimester of gestational age by detecting IgM antibodies.” I advise EXTREME caution when diagnosing acute toxoplasmosis solely on IgM antibodies. PLEASE, research the role of IgA antibodies against T. gondii and IgG avidity in the diagnosis of acute toxoplasmosis and rephrase this paragraph.
Author contributions, funding, data availability, conflict of interest, author contributions: please revise, no need to let the MDPI guidelines or the “ “ in there ….
I consider this article to be important for the field and should be considered for publication if the authors consider fixing the issues i've pointed out, especially the diagnosis of acute toxoplasmosis.
Wishing you the best,
Reviewer
Comments on the Quality of English LanguageThey need to re-check the manuscript and remove some mdpi guidelines and also, translate some spanish i guess, into english. Besides that, nothing to add. English is clear to me.
Author Response
Comments 1: Thank you for the opportunity to review this manuscript.
Response 1: I would like to thank you for all the suggestions I received. I am certain they had a major impact on this work. The suggested changes were made in the article. However, if you prefer to more information at work, we are prepared to modify it. Thank you.
Comments 2: Abstract: please write “T. gondii” in Italic throughout the manuscript.
Introduction: I would personally add a paragraph on serology for the diagnosis of acute/chronic toxoplasmosis.
Response 2: The correction was made. Thank you.
Comments 3: Materials and methods: “2.5 Análise laboratorial” does not sound English to me ☹
Response 3: The correction was made. Thank you.
Comments 4: Please add the assay and the reagent for all the laboratory tests included in the study.
Response 4: The correction was made. Thank you.
Comments 5:
Results:
Line 154: “suggesting recent infection” I would delete this statement without an additional IgA antibody against T. gondii test and an IgG avidity test.
Please see: https://www.ncbi.nlm.nih.gov/pmc/articles/PMC4609698/
Response 5: The correction was made. Thank you.
Comments 6:
I don’t see anywhere in the manuscript a comparison between urban and rural areas, since T. gondii seropositivity is linked to rural areas, and you know the location of each health unit, please include this comparison as well. Or, if for some reason you don’t want to do it, maybe due to the overrepresentation of the urban regions, mention it in the newly limitation section ?.
Response 6: We acknowledge the overrepresentation of urban regions in our study, with only one health unit located in a rural area. Given this imbalance, a direct comparison between urban and rural areas regarding T. gondii seropositivity may not yield Statistically meaningful results. Therefore, we have chosen not to perform this analysis. This limitation has been addressed in the newly added in manuscript (Data Analysis). Thank you.
Comments 7:
Discussion:
Please add the study limitations; maybe not determining IgA against T. gondii and IgG avidity in IgM positive females could be considered one.
Response 7: As suggested, the absence of IgA determination against T. gondii and the lack of IgG avidity testing in IgM-positive females have been recognized as limitations of this study and have been included. Thank you.
Comments 8:
Line 245: “acute infection (IgM) in pregnant women was diagnosed in 20 (6.7%) cases during 245 the first trimester of gestational age by detecting IgM antibodies.” I advise EXTREME caution when diagnosing acute toxoplasmosis solely on IgM antibodies. PLEASE, research the role of IgA antibodies against T. gondii and IgG avidity in the diagnosis of acute toxoplasmosis and rephrase this paragraph.
Response 8: The correction was made. Thank you.
Comments 9:
Author contributions, funding, data availability, conflict of interest, author contributions: please revise, no need to let the MDPI guidelines or the “ “ in there ….
Response 9: The correction was made. Thank you.
Comments 10:
I consider this article to be important for the field and should be considered for publication if the authors consider fixing the issues i've pointed out, especially the diagnosis of acute toxoplasmosis.
Response 10: Thank you for recognizing the importance of our study to the field. We appreciate your valuable feedback and have carefully addressed all the issues you pointed out, particularly regarding the diagnosis of acute toxoplasmosis. The manuscript has been revised accordingly to enhance its clarity and scientific rigor.

Reviewer 2 Report
Comments and Suggestions for Authors
Overall, the manuscript presents valuable research on the seropositivity of Toxoplasma gondii and co-infection with TORCH complex pathogens among pregnant women in Araçatuba, Brazil. The study's findings are significant and contribute to public health knowledge, particularly in areas of prenatal care and congenital infections. However, a few minor revisions are necessary to enhance the clarity and comprehensiveness of the manuscript:
Line 29: It is recommended to add more relevant keywords to improve the manuscript's visibility and accessibility.
Lines 56-57: The discussion on the prevalence of toxoplasmosis would benefit from the inclusion of additional studies, especially high-quality meta-analyses, to provide a more robust background. Additionally, more local prevalence and incidence data should be included, particularly for the area where the study was conducted.
Line 86: The term "on1" requires clarification to ensure readers understand its context and meaning.
Line 106: Ensure that the language used throughout the manuscript is consistent with standard English usage.
Furthermore, it is essential to mention the criteria that were followed to conduct the study and provide details on the anonymization methods used to protect participants' identities.
Author Response
Comments 1: Overall, the manuscript presents valuable research on the seropositivity of Toxoplasma gondii and co-infection with TORCH complex pathogens among pregnant women in Araçatuba, Brazil. The study's findings are significant and contribute to public health knowledge, particularly in areas of prenatal care and congenital infections. However, a few minor revisions are necessary to enhance the clarity and comprehensiveness of the manuscript:
Response 1:
Thank you for your positive feedback on our manuscript. We are pleased to hear that you find our research valuable and relevant to public health, particularly in the context of prenatal care and congenital infections. We believe that our findings on the seropositivity of Toxoplasma gondii and co-infection with TORCH complex pathogens among pregnant women in Araçatuba, Brazil, provide important insights that can help improve prenatal screening and management practices
Comments 2: Line 29: It is recommended to add more relevant keywords to improve the manuscript's visibility and accessibility.
Response 2: The correction was made. Thank you.
Comments 3: Lines 56-57: The discussion on the prevalence of toxoplasmosis would benefit from the inclusion of additional studies, especially high-quality meta-analyses, to provide a more robust background.
Response 3: The correction was made. Thank you.
Comments 4: Additionally, more local prevalence and incidence data should be included, particularly for the area where the study was conducted.
Response 4: We acknowledge the importance of including more local prevalence and incidence data. However, our comprehensive search of available databases did not yield any studies specific to the area where our study was conducted. Thank you.
Comments 5: Line 86: The term "on1" requires clarification to ensure readers understand its context and meaning.
Response 5: The correction was made. Thank you.
Comments 6: Line 106: Ensure that the language used throughout the manuscript is consistent with standard English usage.
Response 6: The correction was made. Thank you.
Comments 7: Furthermore, it is essential to mention the criteria that were followed to conduct the study and provide details on the anonymization methods used to protect participants' identities.
Response 7: Thank you for highlighting these important points. We have now included a detailed description of the criteria followed in conducting the study, as well as the anonymization methods used to protect participants' identities, to ensure compliance with ethical standards and to maintain participant confidentiality.
